# Homeostatic synaptic depression is achieved through a regulated decrease in presynaptic calcium channel abundance

Michael A Gaviño, Kevin J Ford, Santiago Archila, Graeme W Davis*

Department of Biochemistry and Biophysics, University of California, San Francisco, San Francisco, United States

**Abstract** Homeostatic signaling stabilizes synaptic transmission at the neuromuscular junction (NMJ) of *Drosophila*, mice, and human. It is believed that homeostatic signaling at the NMJ is bi-directional and considerable progress has been made identifying mechanisms underlying the homeostatic potentiation of neurotransmitter release. However, very little is understood mechanistically about the opposing process, homeostatic depression, and how bi-directional plasticity is achieved. Here, we show that homeostatic potentiation and depression can be simultaneously induced, demonstrating true bi-directional plasticity. Next, we show that mutations that block homeostatic potentiation do not alter homeostatic depression, demonstrating that these are genetically separable processes. Finally, we show that homeostatic depression is achieved by decreased presynaptic calcium channel abundance and calcium influx, changes that are independent of the presynaptic action potential waveform. Thus, we identify a novel mechanism of homeostatic synaptic plasticity and propose a model that can account for the observed bi-directional, homeostatic control of presynaptic neurotransmitter release.

*For correspondence: graeme.davis@ucsf.edu

## Introduction

Homeostatic signaling systems are believed to interface with the mechanisms of neural plasticity to stabilize neural function throughout the life of an organism (*Marder and Goaillard, 2006*; *Turrigiano, 2011*; *Davis, 2013*). To do so, homeostatic signaling systems require bi-directional control, being able to accurately offset perturbations that persistently increase or decrease the excitation of neurons or muscle. An evolutionarily conserved, bi-directional form of homeostatic signaling is observed at the neuromuscular junction of organisms ranging from *Drosophila* to human (see *Davis, 2013* for review). At the *Drosophila* NMJ, inhibition of postsynaptic glutamate receptor function leads to a compensatory increase in presynaptic neurotransmitter release that precisely restores normal muscle depolarization (*Petersen et al., 1997*; *Davis and Goodman, 1998*; *Davis et al., 1998*; *Frank et al., 2006*; *Davis, 2013*). This is referred to as presynaptic homeostatic potentiation (PHP). An opposing process, presynaptic homeostatic depression (PHD), can be induced by presynaptic overexpression of the vesicular glutamate transporter (*vGlut2*), which causes an increase in the amount of glutamate packaged within individual synaptic vesicles and a corresponding increase in synaptic vesicle diameter (*Daniels et al., 2004*). These larger vesicles produce larger average spontaneous miniature excitatory post-synaptic potentials (mEPSPs). A homeostatic decrease in presynaptic vesicle release, however, maintains evoked EPSP amplitudes at wild type levels despite increased mEPSP amplitude (*Daniels et al., 2004*).

It remains unknown whether these opposing forms of presynaptic, homeostatic modulation, PHP, and PHD, are coordinately controlled to achieve robust, bi-directional control of muscle excitation. Can PHP and PHD be simultaneously induced to adjust presynaptic release? If so, can the expression

**eLife digest** Neurons are organised into networks via junctions called synapses. The arrival of an electrical signal, called an action potential, at a neuronal synapse causes an influx of calcium ions into the cell. This in turn causes packages of chemicals called neurotransmitters that are stored inside the cell to fuse with the neuron's membrane, which releases their contents from the neuron. These molecules bind to the cell on the other side of the synapse (another neuron or a muscle cell), and the action potential can be regenerated.

Most biological systems are maintained within an optimal range, even in the context of a constantly changing external environment. The nervous system is no exception. The communication across synapses can be carefully controlled to ensure that action potentials pass through a neural network in a reliable way. This means that deviations away from an optimal amount of synaptic transmission trigger changes that compensate for the deviation and aim to restore the optimum. When these compensatory changes increase synaptic transmission, the process is referred to as 'homeostatic potentiation'; when they reduce it, this is known as 'homeostatic depression'.

Gaviño et al. studied the synapses between neurons and muscle cells in the larvae of fruit flies that had been genetically modified to produce abnormally large packages of neurotransmitters. In a clear demonstration of homeostatic depression, the larvae compensate for their altered condition by releasing fewer of their extra large packages in response to each action potential. Mutations that disrupt homeostatic potentiation have no effect on this process, suggesting that the two processes work via separate mechanisms. Indeed, further experiments reveal that synapses achieve homeostatic depression by reducing the number of calcium channels in the membrane. This limits the entry of calcium—and thus the release of neurotransmitters—in response to each action potential.

Homeostatic potentiation and depression are independent processes with distinct mechanisms, and which work in combination to allow fine-grained control of communication across synapses. Future work is now needed to determine how the mechanisms of homeostatic depression and potentiation are coordinated to precisely control neural activity, and if these mechanisms are also commonly employed at synapses in the mammalian brain.

of PHP and PHD be coordinated to produce an accurate homeostatic response, precisely offsetting the magnitude of the perturbation? Are PHP and PHD controlled by the same signaling system(s) or are they independently controlled and coordinated? Ultimately, understanding how these signaling systems interface requires the identification and characterization of the cellular and molecular mechanisms that drive PHP and PHD. Although significant progress has been made in identifying the mechanisms that participate in PHP (*Frank et al., 2006*; *Dickman and Davis, 2009*; *Tsurudome et al., 2010*; *Müller et al., 2011*; *Müller and Davis, 2012*; *Younger et al., 2013*), nothing is known regarding the cellular or molecular mechanisms responsible for PHD (*Daniels et al., 2004*).

Here, we demonstrate that PHP and PHD are driven by distinct molecular mechanisms. Further, we demonstrate that PHD is achieved by a regulated decrease in presynaptic calcium channel abundance and concomitant decrease in presynaptic calcium influx. The regulated control of active-zone associated calcium channel abundance represents a novel mechanism of homeostatic synaptic plasticity. Importantly, this mechanism is consistent with the simultaneous induction of both PHP and PHD. As such, bi-directional, rheostat-like adjustment of presynaptic release could reasonably be achieved by the coordinated expression of two independent homeostatic signaling systems at the NMJ.

## Results

### Presynaptic homeostatic plasticity is bi-directional

We first confirmed that presynaptic overexpression of the *vGlut2* glutamate transporter (*UAS-vGlut2*) caused an increase in average mEPSP amplitude and a corresponding decrease in presynaptic vesicle release (*Daniels et al., 2004*; *Figure 1A–E*). Animals overexpressing *UAS-vGlut2* (referred to hereafter as *vGlut*-OE animals) displayed a shift of the entire mEPSP amplitude distribution, corresponding to larger miniature release events (*Figure 1E*). As previously shown (*Daniels et al., 2004*),

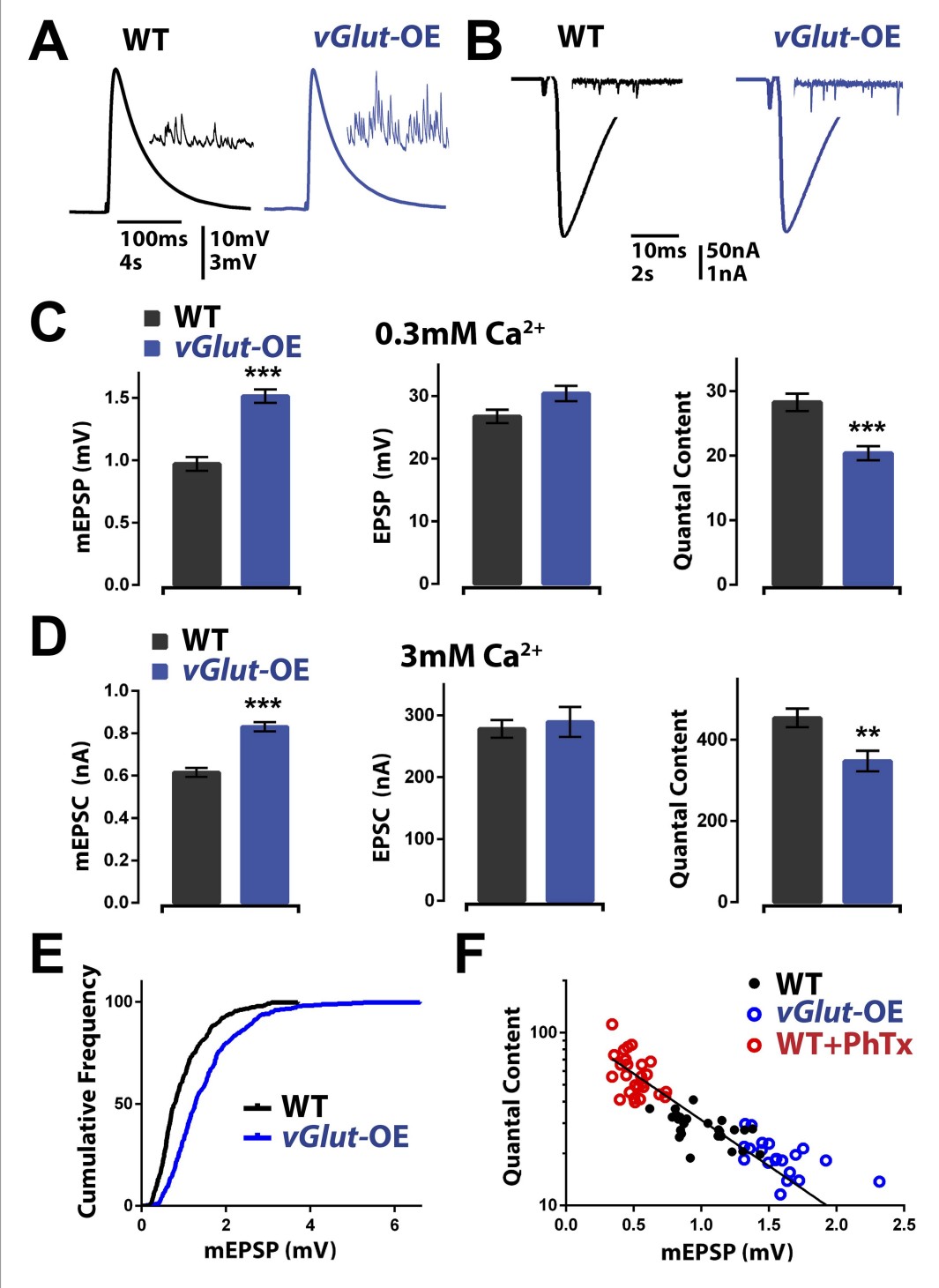

**Figure 1**. *vGlut*-OE animals display increased miniature EPSP amplitude and a compensatory decrease in vesicle release. (**A**) Representative EPSP traces following stimulation of a single action potential in wild type and *vGlut*-OE animals; representative mEPSP traces in inset. (**B**) Representative EPSC traces following stimulation of a single action potential in wild type and *vGlut*-OE animals; representative mEPSC traces in inset. (**C**) *vGlut*-OE animals have increased mEPSP amplitudes (p < 0.0001) with no change in evoked EPSP amplitudes (p = 0.08), resulting in a decrease in calculated quantal content (p < 0.0001, n > 12 NMJ per genotype). (**D**) *vGlut* overexpression results in increased mEPSC amplitudes (p < 0.0001), but no significant change in evoked EPSC amplitudes, resulting in a decrease in calculated quantal content (p < 0.01, n > 8 per genotype, error bars = SEM). (**E**) The cumulative

*Figure 1. Continued*

frequency of mEPSPs is shifted to greater amplitudes in *vGlut*-OE animals. (**F**) Quantal content is able to scale across an order of magnitude in response to perturbations that increase (*vGlut2* overexpression) or decrease (PhTx treatment) quantal amplitude. An exponential function was fit to these data points with $R^2 = 0.7491$. $[Ca]^e = 3$ mM for (**B**) and (**D**), 0.3 mM for all others. Error bars = SEM for all.

a decrease in presynaptic release is able to precisely offset the increase in mEPSP amplitude, thereby restoring EPSP amplitudes to wild type levels (*Figure 1A,C*). Since EPSP amplitudes are maintained at wild type levels, these data support the conclusion that a homeostatic signaling system detects the change in average mEPSP amplitude and drives a compensatory reduction in presynaptic neurotransmitter release that precisely offsets the magnitude of increased mEPSP amplitude. This compensatory response will be referred to as PHD to distinguish this process from PHP.

Following the induction of PHP, there is a rightward shift in the calcium cooperativity of neurotransmitter release, indicating that the homeostatic potentiation of neurotransmitter release is robust over a range of extracellular $Ca^{2+}$ concentrations and likely does not involve a change in the calcium sensor for synaptic vesicle fusion (*Frank et al., 2006*; *Dickman and Davis, 2009*; *Müller et al., 2012*). We performed a similar analysis for PHD in *vGlut*-OE animals. We observed that PHD accurately offset an increase in average spontaneous release event amplitude at elevated extracellular calcium (3 mM; *Figure 1B,D*), demonstrating that PHD proceeds accurately at extracellular calcium concentrations $[Ca^{2+}]_e$ that span what is considered to be physiological in the intact organism (compare quantal content in *Figure 1C,D*). Experiments at elevated extracellular calcium, 3 mM $[Ca^{2+}]_e$, were performed using two-electrode voltage clamp (see 'Materials and methods'). Similar changes were observed at a range of intervening calcium concentrations (0.3–1.5 mM) with no change in the slope of the calcium cooperativity curve, consistent with previous experiments examining calcium cooperativity in *vGlut*-OE animals over a narrower range of extracellular calcium (data not shown; *Daniels et al., 2004*). These data demonstrate that PHD is robust across a range of extracellular calcium concentrations and is likely to be independent of any change in the calcium sensor for presynaptic release.

We next addressed the accuracy of homeostatic signaling during PHD. A system that is under true homeostatic control should be able to accurately offset a wide range of perturbation. This was previously shown for PHP by plotting the average mEPSP amplitude against average quantal content for each recording made from an individual NMJ. In this analysis, the data lie along a line associated with near perfect compensation (*Frank et al., 2006*). We constructed a similar plot, including recordings from *vGlut*-OE NMJ as well as wild type NMJ and NMJ that were incubated in sub-blocking concentrations of the glutamate receptor antagonist philanthotoxin-433 (PhTx), and observed a wide range of mEPSP amplitudes, ranging from >2 mV to less than 0.4 mV. These data produced points lying along a clear line, providing evidence for continuous adjustment of presynaptic release, effectively maintaining a constant EPSP amplitude over a wide range of average mEPSP amplitude values (*Figure 1F*). Moreover, each segment of the data in *Figure 1F* (shown by different colors), when fit independently, showed a negative relationship between mEPSP amplitude and presynaptic release. This indicates that, as with PHP, the mechanisms of PHD can respond to small changes in the average mEPSP amplitude and adjust presynaptic release accordingly. In conclusion, these data suggest that PHD could be part of an integrated, bi-directional, homeostatic signaling system at the NMJ.

## PHP and PHD can be combined to fine tune presynaptic release at a single NMJ

We next asked whether the NMJ can coordinate simultaneous induction of homeostatic potentiation and depression to precisely control release. Since *vGlut2* overexpression is a presynaptic perturbation and PhTx is a postsynaptic perturbation, both PHD and PHP can be induced concurrently. First, we found that PhTx application to *vGlut*-OE terminals significantly reduced mEPSP amplitudes in a manner quantitatively similar to that observed when PhTx is applied to wild type NMJ (*Figure 2A,B*). At both wild type and *vGlut*-OE NMJ, the 50–60% drop in average mEPSP amplitude was offset by an

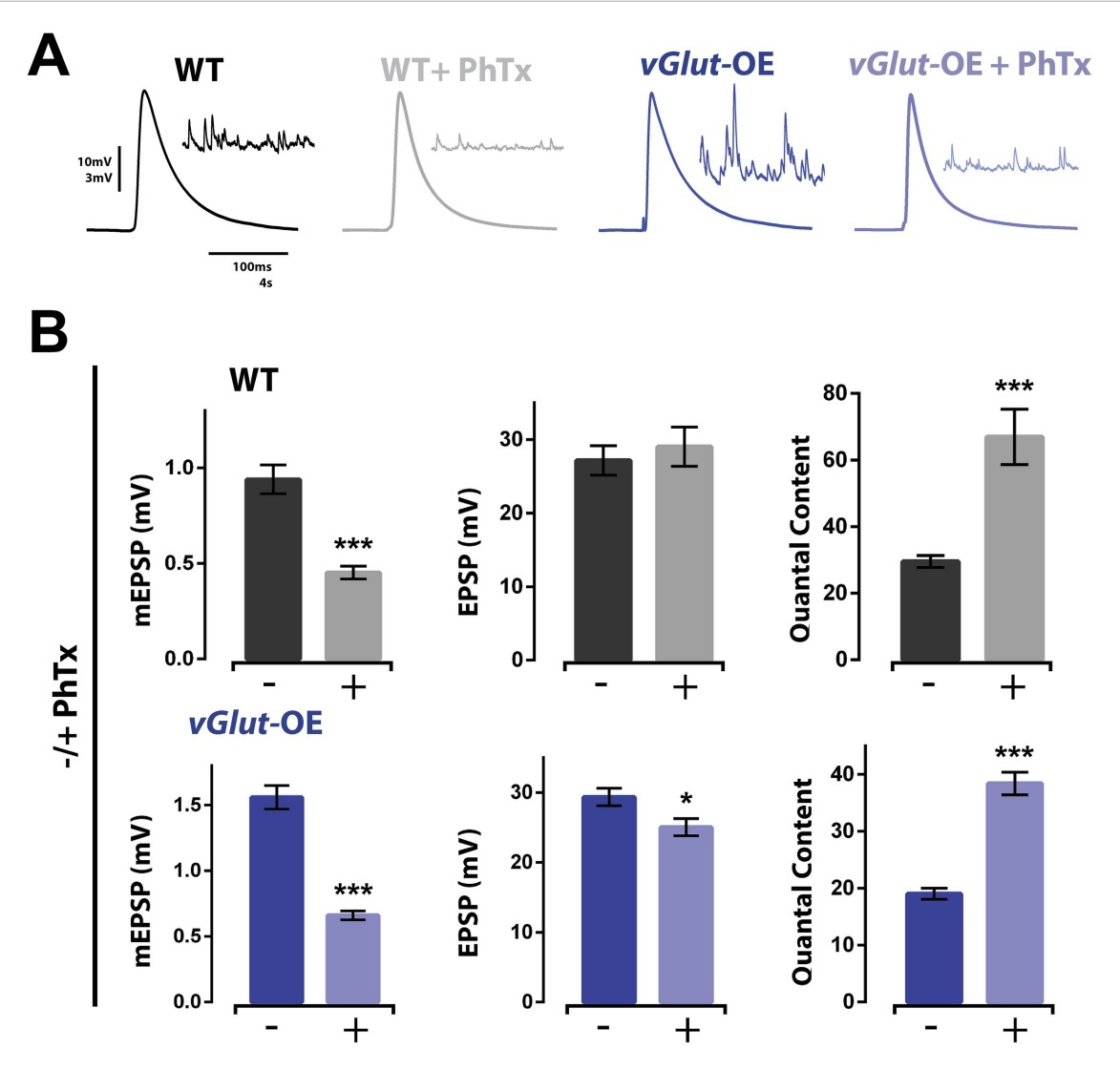

**Figure 2**. Homeostatic potentiation can be induced at NMJ already expressing presynaptic homeostatic depression (PHD). (**A**) Representative evoked EPSP traces from wild type and *vGlut*-OE NMJ without and with the application of PhTx. Corresponding mEPSP traces in inset for each condition. (**B**) The addition of PhTx to either wild type (top) or *vGlut*-OE (bottom) animals results in decreased mEPSP amplitude (p < 0.0001 for both) with a minimal change in evoked EPSP amplitude (p = 0.5 for wt, p = 0.03 for *vGlut*-OE), resulting in a decrease in calculated quantal content (p < 0.0001 for both, n = 11 for wt -PhTx, n = 8 for wt +PhTx, n = 6 for *vGlut*-OE–PhTx, n = 9 for *vGlut*-OE +PhTx. Error bars = SEM). [Ca]ᵉ = 0.3 mM for all.

approximate doubling of presynaptic neurotransmitter release (*Figure 2A,B*). As a result, in both wild type and *vGlut*-OE NMJ, EPSP amplitudes were maintained at or near wild type levels after PhTx application (a slight decrease in EPSP amplitude was observed in *vGlut*-OE animals, p < 0.05). Furthermore, there was no difference in EPSP amplitude comparing wild type (+PhTx) and *vGlut*-OE (+PhTx; p > 0.05). Thus, despite the fact that *vGlut*-OE NMJ chronically express PHD during larval development, application of PhTx is able to induce expression of PHP that appears unaffected by prior induction of PHD. From these results, several conclusions can be made. First, these data demonstrate that the mechanisms of PHP and PHD can co-exist at a single NMJ and can be sequentially induced, at least in the sequence that we performed in this study. The reagents to perform the reverse sequence do not currently exist as there is no pharmacological or acute genetic manipulation that can rapidly increase mEPSP amplitude. Second, the data argue that the reduced presynaptic release caused by *vGlut2* overexpression is not a secondary effect whereby *vGlut2*

overabundance disrupts the vesicle fusion apparatus, since release can be precisely potentiated at this NMJ. Finally, these data demonstrate that the mechanisms of PHP and PHD can be effectively combined to fine tune presynaptic release.

## Distinct molecular mechanisms drive PHP and PHD

Since PHP and PHD can be combined to produce an accurate homeostatic response, and since both PHP and PHD act upon the presynaptic release of neurotransmitter, it raises the question as to whether PHP and PHD are controlled by the same molecular mechanism. In one scenario, the same signaling system(s) could drive both PHP and PHD. Alternatively, distinct molecular mechanisms might underlie PHP and PHD and the two processes might somehow be combined or coordinated to produce an accurate homeostatic response. Considerable progress has been made identifying genes that, when mutated, block PHP. Therefore, we tested several of these gene mutations for a role in PHD.

It was recently demonstrated that the potentiation of neurotransmitter release during PHP requires a presynaptic ENaC channel (*Younger et al., 2013*). Briefly, in either a glutamate receptor mutant background, or following acute application of PhTx to the NMJ, subsequent pharmacological inhibition of the presynaptic ENaC channel, or genetic mutation of the gene encoding an essential subunit of this channel, eliminates the expression of homeostatic potentiation (*Younger et al., 2013*). A current model suggests that sodium leak through newly membrane-inserted ENaC channels drives presynaptic membrane depolarization, a subsequent potentiation of presynaptic calcium influx, and enhanced neurotransmitter release. Importantly, ENaC channels do not appear to be present on the membrane under baseline conditions since deletion or pharmacological blockade of these channels does not decrease baseline neurotransmitter release (*Younger et al., 2013*). If ENaC channels are not present on the membrane at rest, then these channels should not have a role in PHD. To test this possibility, we applied the ENaC channel inhibitor benzamil (25 µM) to wild type and *vGlut*-OE NMJ. We found no significant effect of benzamil on presynaptic release at *vGlut*-OE NMJ, consistent with the expectation that ENaC channels do not participate in baseline release in wild type animals and, therefore, cannot be removed as a mechanism underlying PHD. (*Figure 3A,B*). As a control, we tested whether application of Benzamil still prevents PHP expression in the *vGlut*-OE background. We found that this is, indeed, the case (*vGlut*-OE +Benzamil: mEPSP = 0.83 ± 0.05 mV, QC = 39.2 ± 2.3; *vGlut*-OE +Benzamil +PhTx: mEPSP = 0.50 ± 0.02 mV, QC = 43.6 ± 3.0; quantal content is statistically unchanged; p = 0.29, n > 10). Taken together, these data imply that the mechanisms responsible for PHD are molecularly separable from those that drive PHP.

To further investigate molecular distinctions between PHP and PHD, we examined additional gene mutations shown to block PHP. We first tested whether mutations in the gene encoding Rab3 interacting molecule (RIM) affect homeostatic synaptic depression. We previously demonstrated that multiple loss of function alleles of the *rim* gene, including a deletion that removes a large portion of the coding sequence (*rim*[103]), all block PHP, independent of whether they also impair baseline presynaptic release (*Müller et al., 2012*). By contrast, *vGlut2* overexpression in the *rim*[103] mutant background still induced robust expression of PHD (*Figure 3C,D*). This experiment was performed in 0.4 mM extracellular calcium to normalize presynaptic calcium influx and vesicle release to that observed in wild type under standard recording conditions (*Müller et al., 2012*). These data demonstrate that RIM is not required for PHD and further separate the mechanisms that promote PHD vs PHP.

Finally, we tested a point mutation in the CaV2.1 calcium channel previously shown to block PHP (*cac*[S]; *Frank et al., 2006*). The *cac*[S] mutation is a single amino acid substitution in the sixth transmembrane domain of the third repeat of the pore forming subunit of the CaV2.1 channel (*Brooks et al., 2003*). In a current model of PHP, the *cac*[S] mutation prevents low-voltage modulation of CaV2.1 calcium channels following insertion of ENaC channels in the presynaptic membrane (*Younger et al., 2013*). Because the *cac*[S] mutation causes a ~30% decrease in presynaptic calcium influx and a corresponding decrease in presynaptic release at baseline (*Müller and Davis, 2012*), we performed our analysis at elevated (1 mM) extracellular calcium. By elevating extracellular calcium, we restored EPSP amplitudes and calcium influx to levels observed in wild type animals under standard experimental conditions (0.3 mM extracellular calcium). Although the *cac*[S] mutation completely blocks PHP, we observed no effect on PHD at *vGlut*-OE NMJ (*Figure 3E,F*). Taken together with the observation that PHD is insensitive to benzamil treatment and occurs normally in the presence of

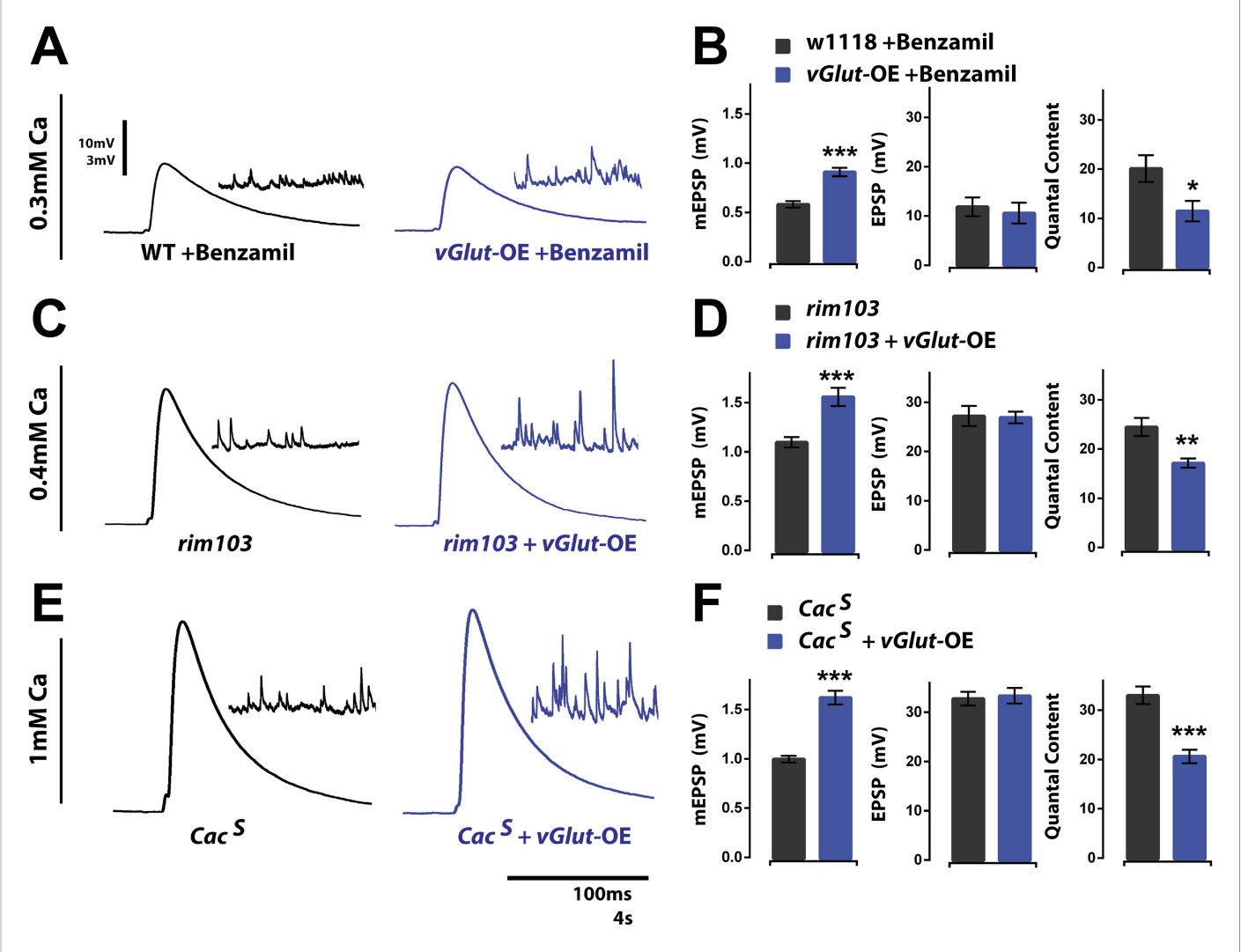

**Figure 3**. Presynaptic homeostatic potentiation and depression operate through distinct molecular mechanisms. (**A**) Representative single action potential stimulated EPSP traces from wild type and *vGlut*-OE animals in the presence of the drug Benzamil with corresponding mEPSP traces in the inset. (**B**) *vGlut*-OE animals have increased mEPSP amplitudes in the presence of Benzamil as compared to similarly treated wild type animals (p < 0.0001) but have unchanged evoked EPSP amplitudes, resulting in a decrease in calculated quantal content (p = 0.05, n = 9 for wt, n = 5 for *vGlut*-OE). (**C**) Representative EPSP traces as in (**A**) from *rim103* homozygotes in a wild type (left) and *vGlut2* overexpressing (right) background. (**D**) *vGlut2* overexpression in *rim103* mutants results in increased mEPSP amplitudes (p = 0.0002) and decreased quantal content (p = 0.003, n = 12 for *rim103* and n = 10 for *rim103* +*vGlut*-OE). (**E**) Representative EPSP traces as in (**A**) from *Cac^S* homozygotes in a wild type (left) and *vGlut2* overexpressing (right) background. (**F**) *vGlut2* overexpression in *Cac^S* mutants results in increased mEPSP amplitudes (p < 0.0001) and decreased quantal content (p < 0.0001, n = 22 for *Cac^S*, n = 11 for *Cac^S* +*vGlut*-OE). Error bars = SEM for all.

a RIM deletion, our data demonstrate that the molecular mechanisms underlying PHD are distinct from those that drive PHP.

## Action potential properties are unchanged during presynaptic homeostasis

The expression of PHP is achieved, in part, by an increase in presynaptic calcium influx through CaV2.1 calcium channels (*Frank et al., 2006*; *Müller and Davis, 2012*). In principle, this could be achieved by broadening the waveform of the presynaptic action potential. Conversely, PHD could be achieved by narrowing the presynaptic action potential waveform. A direct test of these possibilities has been

lacking because it has not been possible to record from the type 1 presynaptic boutons of the *Drosophila* NMJ, which are embedded within the postsynaptic muscle cell. However, it was recently demonstrated that Archaerhodopsin can be expressed in motoneurons without affecting baseline neurotransmission. The voltage sensitive properties of Archaerhodopsin can be used to visualize the presynaptic action potential waveform with high temporal resolution (*Ford and Davis, 2014*). Importantly, this technique has sufficient temporal resolution to quantify changes in action potential half-width that correlate with either a 50% increase or decrease in neurotransmitter release (*Ford and Davis, 2014*). Therefore, this tool has sufficient sensitivity to resolve changes in action potential waveform that might be responsible for PHD or PHP.

To determine whether PHP and/or PHD are achieved by a change in action potential waveform, we expressed Archaerhodopsin (Arch-GFP) in motoneurons of wild type animals with and without PhTx treatment, in *GluRIIA* mutant animals, and in *vGlut*-OE animals. This level of Arch-GFP expression does not alter baseline synapse function or anatomy (*Ford and Davis, 2014*). To image voltage, we performed confocal spot measurements, using GFP to localize the spot to the edges of individual synaptic boutons. Importantly, spot confocal imaging does not alter action potential propagation past the imaging site to a distal extracellular recording site within an NMJ and spot excitation at 643 nm has no effect on baseline neurotransmission (*Ford and Davis, 2014*). It is worth noting that measurements of action potential waveform at the NMJ are nearly identical to measurements made from somatic patch clamp recordings at the motoneuron cell body (*Marie et al., 2010*; *Ford and Davis, 2014*).

Using Arch-GFP to image the presynaptic AP waveform, we observed that the induction of PHP following application of PhTx to the NMJ or in animals containing a mutation in the muscle specific glutamate receptor *GluRIIA* did not cause an increase in AP width or WHM (*Figure 4A–C*). We found a small, but significant, decrease in WHM following PhTx application, but this is in the opposite direction expected for the potentiation of neurotransmitter release (*Figure 4C*). Next, we observed the AP waveform in *vGlut*-OE animals. This experiment was performed at two different extracellular calcium concentrations: low extracellular calcium (0.4 mM) typically used to examine synaptic transmission in current clamp configuration, and at physiological calcium (1.5 mM) to control for the recent observation that action potential repolarization is faster at elevated extracellular calcium (*Figure 4A*, far right, compare black and gray traces; *Ford and Davis, 2014*). In both cases, there was no change in AP waveform comparing wild type with *vGlut*-OE animals (*Figure 4A–C*). We also controlled for the possibility that Arch-GFP imaging might alter the expression of PHP or PHD by recording neuromuscular transmission from the same genotypes in which we quantified AP waveforms. We found that both PHP, using either PhTx or the *GluRIIA* mutation, and PHD occur normally in the presence of Arch-GFP (*Figure 4D,E*). From these data, we conclude that the homeostatic modulation of presynaptic neurotransmitter release, either PHP or PHD, is not correlated with a change in AP waveform measured at the presynaptic nerve terminal.

## Homeostatic depression is not correlated with a change in the RRP

PHP requires two parallel changes within the presynaptic nerve terminal; (1) an increase in calcium influx through CaV2.1 calcium channels and (2) an increase in the RRP. If homeostatic depression is controlled by similar presynaptic mechanisms, then we would expect the process to be correlated with decreased presynaptic calcium influx and decreased RRP size. We therefore measured these parameters in *vGlut*-OE animals.

First, we assayed the RRP in *vGlut*-OE animals by quantifying EPSC amplitudes during high frequency stimulation (30 stimuli at 60 Hz, 3 mM extracellular calcium) and estimating the cumulative EPSC and RRP according to published methods (see 'Materials and methods'). Estimates of the RRP using this method in wild type animals agree with estimates made according to variance mean analysis, a technique based on the measurement of EPSC amplitude variance delivered at low (0.2 Hz) stimulus frequencies that do not induce short term synaptic modulation (*Müller et al., 2012*). We found that the RRP was not significantly different when comparing wild type and *vGlut*-OE animals (*Figure 5A,B*). Based on these data, we conclude that homeostatic depression is not correlated with a drop in the size of the RRP, highlighting a mechanistic difference in the expression of PHP and PHD.

Having observed no change in the RRP size of *vGlut*-OE animals, we assessed whether PHD is associated with a drop in presynaptic release probability ($P_r$) by examining short-term synapse modulation. As a measure of $P_r$, we calculated the fraction of the total RRP released following the first

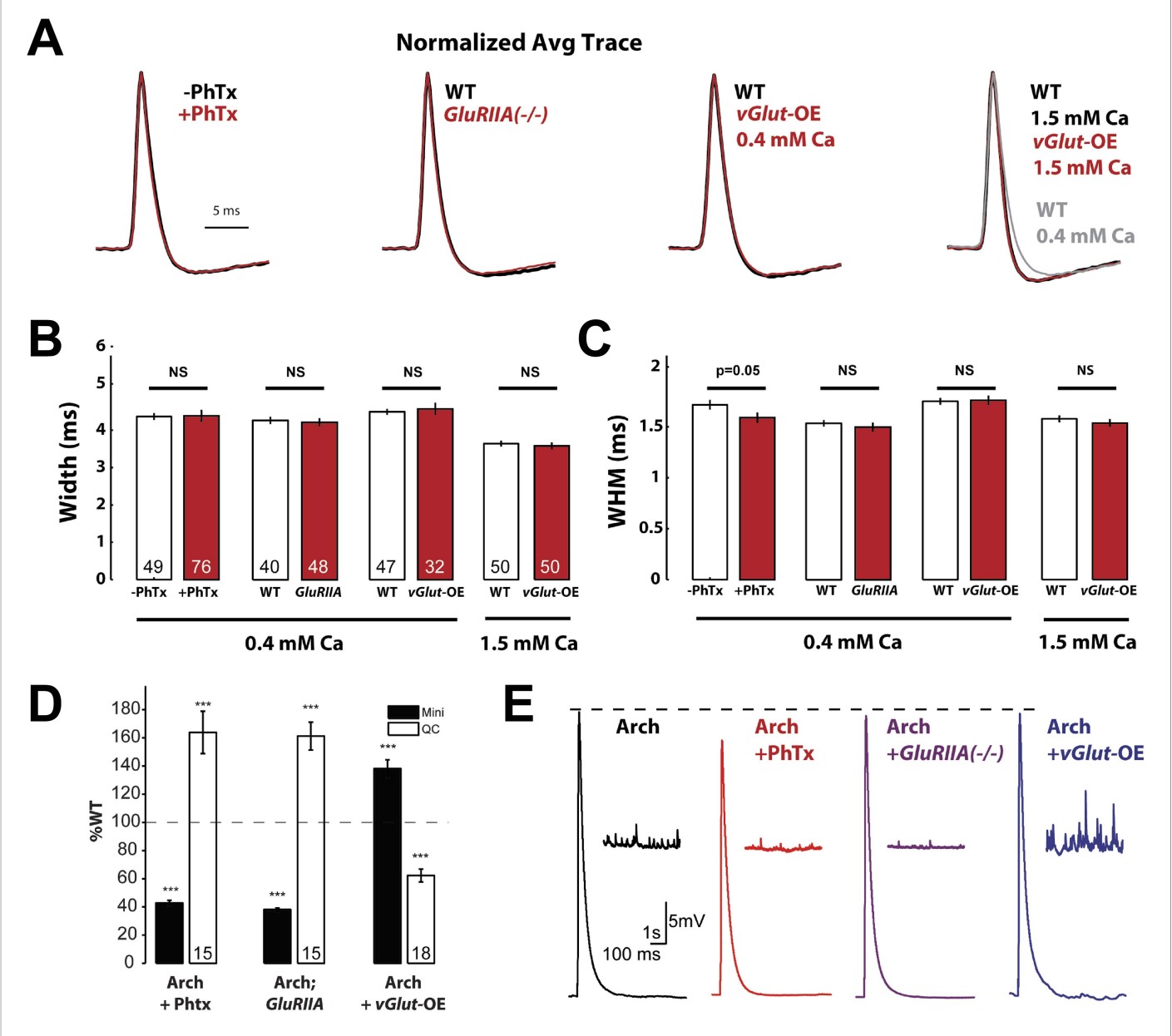

**Figure 4.** Action potential waveforms do not change during presynaptic homeostasis. (**A**) Normalized average AP traces comparing PhTx-treated, *GluRIIA* mutant, and *vGlut*-OE animals to WT animals. (**B**) AP width does not change in animals in which presynaptic homeostasis has been induced (n labeled within bars for all). (**C**) AP width at half maximum amplitude does not change in animals where presynaptic homeostasis has been induced, except for a small decrease in PhTx treated animals; p = 0.05; n for all is same as (**B**). (**D**) Quantal content homeostatically changes in response to perturbed mEPSP amplitude in animals that express Arch (p < 0.001, n labeled within bars for all). (**E**) Example traces of the data summarized in (**D**). $[Ca]^e$ = 0.4 mM, errors bars = SEM for all.

action potential in a stimulus train, a value referred to as $P_{train}$ (*Schneggenburger et al., 1999*). Using this method, we found $P_{train}$ to be significantly lower in *vGlut*-OE animals as compared to wild type, indicating reduced presynaptic release probability (*Figure 5C*). Examining accumulated synaptic depression during a short stimulus train, however, we observed only a mild, not statistically significant, difference in the rate of depression (*Figure 5D*). Finally, we assessed paired pulse stimulation at low extracellular calcium (0.3 mM) at a range of inter-stimulus intervals and found no change in paired-pulse depression (*Figure 5E*). In summary, we observed a significant reduction in $P_{train}$, but this

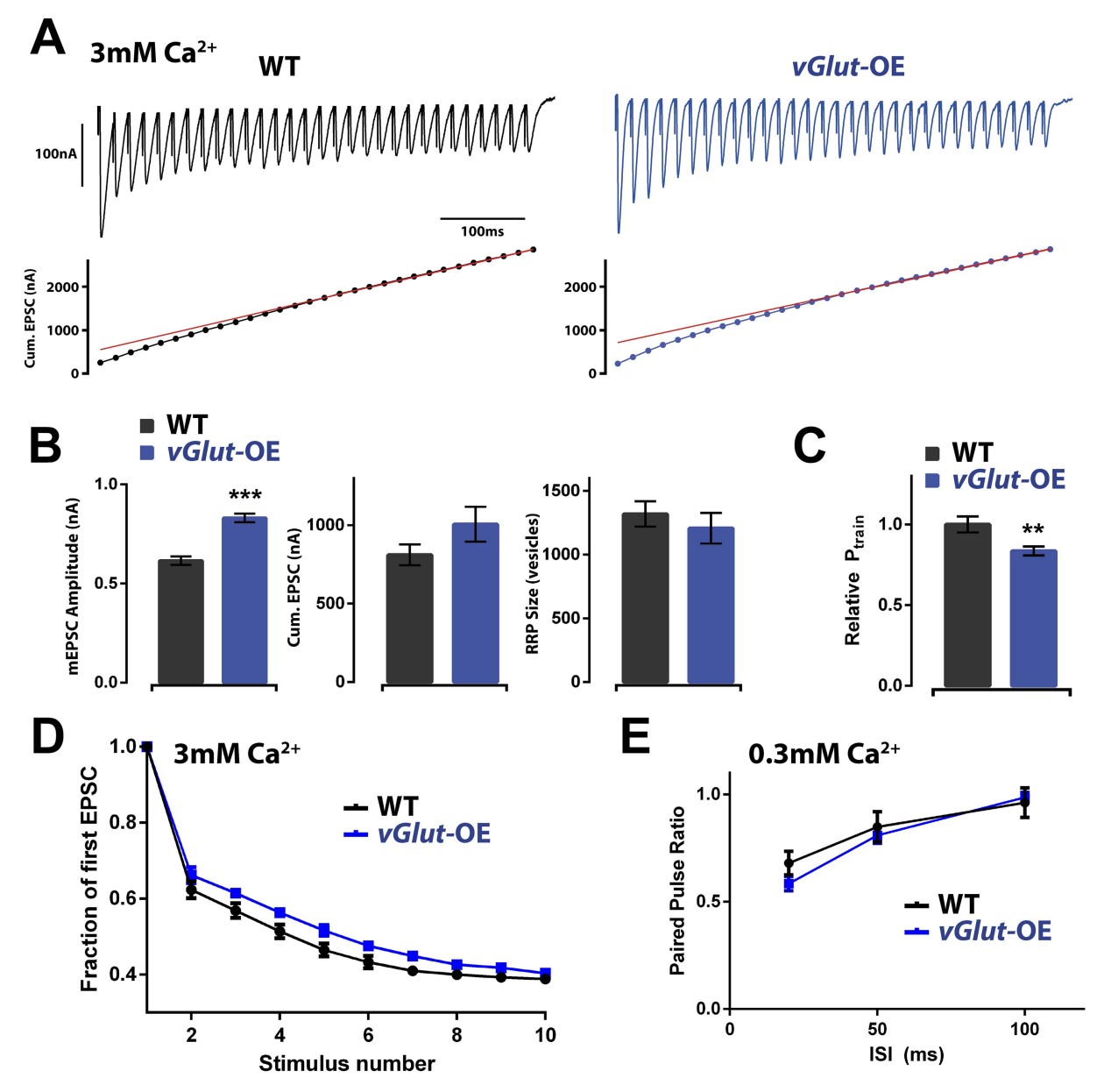

**Figure 5**. Effects of PHD on RRP size and $P_r$. (**A**) Top: representative EPSC trains from wild type (left) and *vGlut*-OE (right) animals in response to 60 Hz stimulation (30 stimuli). Bottom: cumulative EPSC amplitude for the traces shown. The line fit to the cumulative EPSC data and back extrapolated to time 0 is shown in red (see 'Materials and methods'). (**B**) *vGlut*-OE animals had increased mEPSC amplitudes ($p < 0.0001$), and cumulative EPSC amplitudes that trended upward ($p = 0.16$). The calculated RRP size was unchanged ($p = 0.49$). (**C**) The relative probability of release as calculated by the train method was decreased in *vGlut*-OE animals at 3 mM external $Ca^{2+}$ ($p = 0.008$). (**D**) *vGlut*-OE animals were not significantly more resistant to depression over the first 10 stimuli of the 60 Hz trains described (a single phase decay function was fitted to points 2–10 for each muscle, generating a decay constant which was averaged across each genotype; $p = 0.24$). (**E**) The probability of release was unchanged in *vGlut*-OE animals as measured by the paired pulse ratio at 0.3 mM external $Ca^{2+}$ ($n = 5$ for *vGlut*-OE, $n = 6$ for wt). For (**A**) through (**D**): $n = 9$ for WT, $n = 10$ for *vGlut*-OE and $[Ca]^e = 3$ mM. Error bars = SEM for all.

change was not reflected by a change in short-term release dynamics as measured with paired-pulse stimulation at low extracellular calcium. Thus, while pool usage is altered under conditions of high release during a stimulus train, the pool of vesicles is apparently large enough to obscure a change in paired-pulse release dynamics under conditions of low release. We conclude that *vGlut*-OE is associated with a decrease in presynaptic release probability.

# Homeostatic depression is correlated with a decrease in presynaptic calcium influx and channel number

The mechanisms driving PHP include a significant increase in presynaptic calcium influx (*Müller and Davis, 2012*). Therefore, we asked whether there is an opposing decrease in presynaptic calcium during PHD. We estimated presynaptic calcium influx in wild type and *vGlut*-OE animals according to previously published methods (*Müller and Davis, 2012*). In brief, nerve terminals are loaded with the calcium indicator Oregon Green Bapta 1 (OGB1) and a calcium-insensitive reference dye (Alexa 568). Line scans across a single synaptic bouton are then made prior to and following single action potential stimulation. Following single action potential stimulation, we observed a ~50% reduction in calcium influx in *vGlut*-OE animals compared to wild type (*Figure 6A–C*). There was no significant difference in the basal calcium signal, indicating that this effect was not caused by differences in dye loading (*Figure 6C*, left). This decrease in spatially averaged presynaptic calcium influx is more than sufficient to account for the depression of presynaptic neurotransmitter release caused by *vGlut2* over-expression. Indeed, although the precise relationship between the spatially averaged calcium signal and release probability remains to be defined, the drop in the presynaptic calcium signal is larger than might be expected for the observed decrease in presynaptic release observed during PHD (see 'Discussion'). These data indicate that PHD is correlated with a large decrease in presynaptic calcium influx and suggests that this is a primary mechanism driving the process.

A drop in presynaptic calcium influx could be achieved by a change in calcium channel number or function. Currently, no antibodies are available that robustly label the presynaptic calcium channel in vivo. Therefore, we overexpressed a GFP-tagged CaV2.1 channel (referred to, hereafter, as CaV2.1-GFP or Cac-GFP) presynaptically to visualize active-zone associated calcium channels. Over-expression of CaV2.1-GFP does not influence baseline neurotransmission and serves as a reliable reporter of active-zone localized presynaptic channels (*Kawasaki et al., 2004*; *Wang et al., 2014*). Previous reports demonstrate that over-expressed CaV2.1-GFP can report changes in presynaptic calcium channel abundance (*Wang et al., 2014*).

Consistent with prior reports, we found that overexpression of CaV2.1-GFP did not affect transmission in wild type animals and, furthermore, did not affect PHD induction in *vGlut*-OE animals (*Figure 7A,B*). Imaging of overexpressed presynaptic CaV2.1-GFP revealed discrete puncta residing within active zones in both wild type and *vGlut*-OE animals (*Figure 7C,D*). In order to measure CaV2.1 abundance, we first labeled presynaptic active zones with an anti-body against the active zone-associated protein Bruchpilot (Brp), which has previously been shown to surround presynaptic calcium channels based on super-resolution microscopy (*Liu et al., 2011*). We then labeled CaV2.1-GFP with an antibody against GFP and measured the fluorescence intensity of the signal at active zones, using Brp puncta to define the ROI. Strikingly, we observed a 30% and 50% decrease in CaV2.1-GFP signal at type 1B and type 1S boutons, respectively, in *vGlut*-OE animals compared to wild type (*Figure 7E–H*). There was no change in the levels of Brp, assayed by measurement of anti-BRP fluorescence intensity (*Figure 7E–H*).

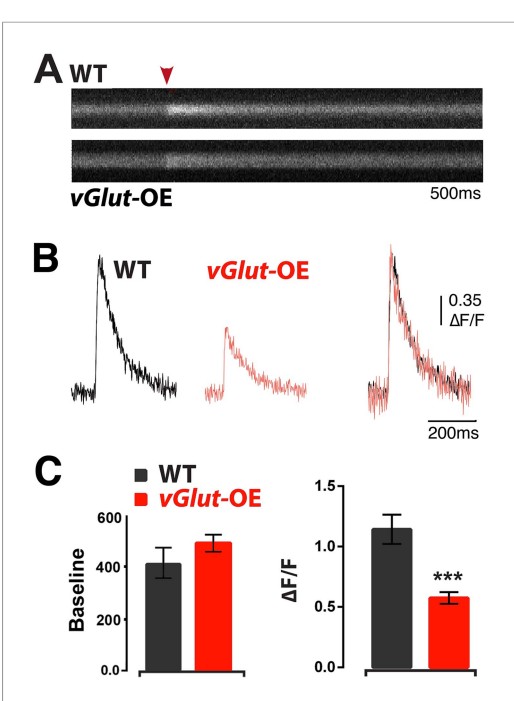

**Figure 6**. Animals expressing PHD have reduced presynaptic calcium influx. (**A**) Representative line-scans of WT and *vGlut*-OE boutons. Red arrowhead: moment of stimulation. (**B**) Example Ca$^{2+}$ transients from WT and *vGlut*-OE animals. *vGlut*-OE transients are identical to WT when normalized by amplitude (right). (**C**) *vGlut*-OE animals displayed a ~50% drop in ΔF/F (right) with no change in baseline OGB-1 fluorescence (left). n = 12 for WT, n = 19 for *vGlut*-OE, p < 0.0001, see 'Materials and methods' for details.

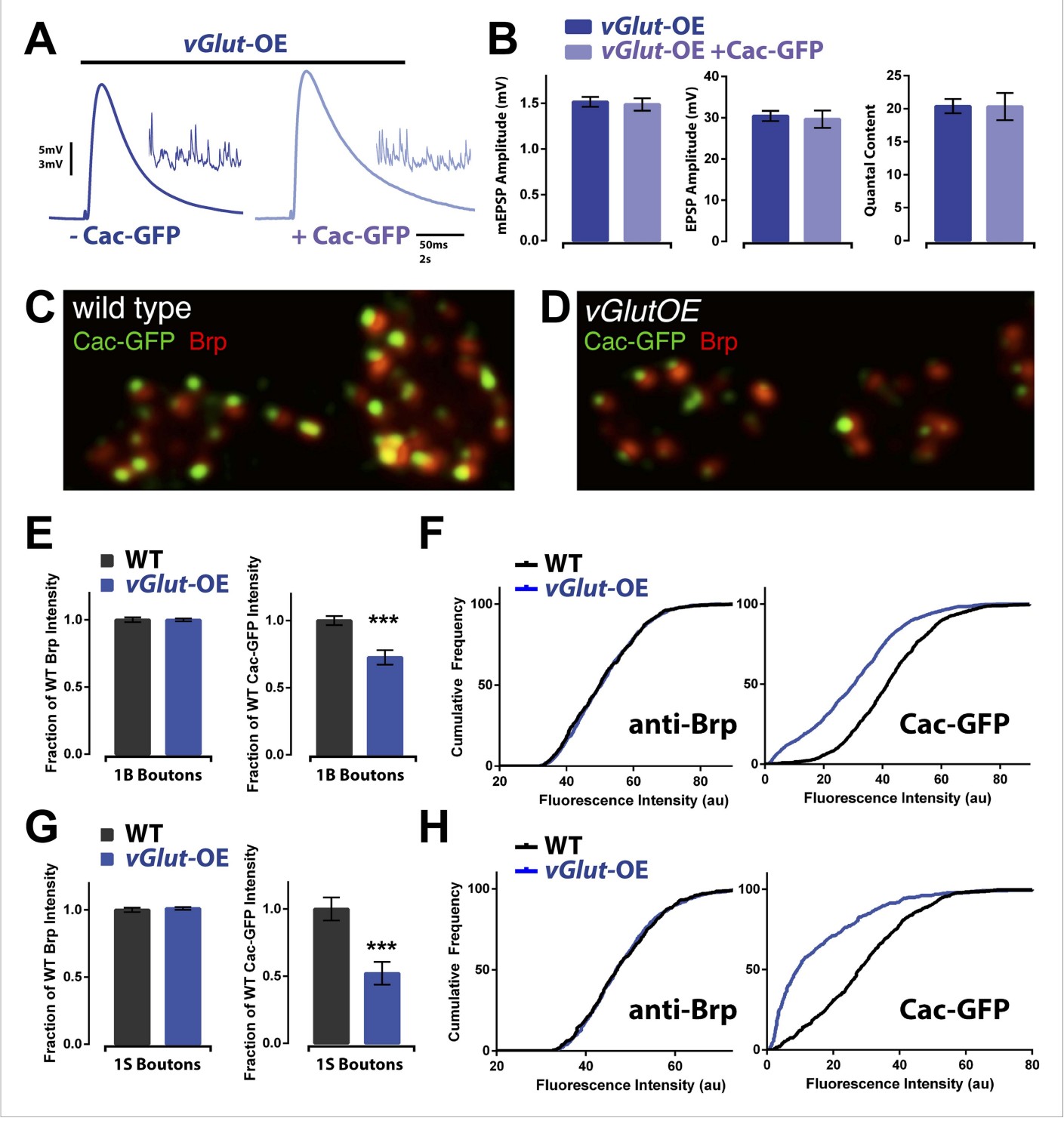

**Figure 7**. Animals expressing PHD have reduced CaV2.1 channel levels. (**A**) Example EPSP traces of *vGlut*-OE animals with (right) and without (left) a *Cac-GFP* transgene; example mEPSPs in inset. (**B**) *vGlut*-OE animals expressing a *Cac-GFP* transgene had unchanged synaptic transmission (n = 15 for *vGlut*-OE, n = 6 for *vGlut*-OE +Cac-GFP, [Ca]$^e$ = 0.3 mM). (**C**) Representative image of a wild type NMJ labeled with antibodies against the active zone component Brp and a transgenically expressed GFP-tagged Cac channel. (**D**) Representative *vGlut*-OE NMJ labeled as in (**C**). (**E**) *vGlut*-OE animals displayed no change in synaptic Brp signal as compared to wild type, but had greatly reduced synaptic Cac-GFP signal at type 1B boutons (p = 0.0002, n > 1000 active zones across >10 NMJ for each). (**F**) The cumulative frequency of fluorescence intensity at 1B boutons for the antibodies described in (**C**). The distribution of *vGlut*-OE animals was shifted to the left as compared to wild type. (**G**) Synaptic Cac-GFP signal was also reduced at type 1S boutons in *vGlut*-OE animals (p = 0.0007, n > 400 active zones across >10 NMJ for each.) (**H**) The cumulative frequency for 1S boutons as described in (**F**). Error bars = SEM for all.

We also quantified the number of Brp puncta per bouton as an estimate of active zone number per bouton and found no significant change (*vGlut*-OE: 12.1 ± 2.0 and 6.4 ± 1.4 puncta per type 1B and 1S bouton, respectively, N = 11 NMJ; WT: 13.3 ± 1.5 and 7.5 ± 2.1. p > 0.05 for both, N = 12 NMJ). These data indicate that the observed decrease in CaV2.1-GFP signal is not a secondary consequence of grossly altered active zone organization, consistent with previous electron microscopic analysis of the *vGlut*-OE NMJ (*Daniels et al., 2004*). Since *UAS-CaV2.1-GFP* expression is independent of the normal transcriptional regulation of the *CaV2.1* gene locus, these data additionally suggest that CaV2.1 levels are not modulated at the level of transcription during PHD, but rather that CaV2.1 protein is selectively restricted from entering the active zone or is selectively removed from the active zone. We conclude that reduced presynaptic CaV2.1 abundance underlies the decreased neurotransmitter release observed during homeostatic depression.

## Discussion

Here, we demonstrate that PHD is driven by a reduction in presynaptic calcium channel abundance at individual active zones. This mechanism is consistent with the co-expression and co-regulation of PHD and PHP, achieving bi-directional homeostatic control of neurotransmission at the NMJ. More specifically, recent data support a model in which PHP is driven by the regulated insertion of an ENaC channel in the presynaptic membrane. This is thought to cause a modest depolarization of the presynaptic membrane, subsequent potentiation of calcium influx and enhanced vesicle release (*Davis, 2013*; *Younger et al., 2013*). This mechanism should function normally, regardless of a change in the number of presynaptic calcium channels per active zone. Therefore, if PHD is driven by a decrease in calcium channel number, PHP can be subsequently induced through insertion of presynaptic ENaC channels. Consistent with this idea, application of an ENaC channel blocker prevents expression of PHP in the *vGlut*-OE background. Although the proposed mechanisms of PHP and PHD are compatible, it is none-the-less remarkable that PHP and PHD can be combined to precisely target the correct EPSP amplitude, offsetting two persistent perturbations: enlarged presynaptic vesicles, and impaired postsynaptic glutamate receptor sensitivity. The mechanisms that enable PHP and PHD to be combined in this manner remain to be elucidated. Our data indicate that the genetic underpinnings responsible for PHP and PHD are distinct, since three conditions that block PHP have no impact on PHD. Thus, we propose that rheostat-like modulation of presynaptic release is achieved by the intersection of two distinct signaling systems driving PHP and PHD.

### Regulating calcium channel number

The regulated trafficking of calcium channels remains a poorly understood process. This is due, in part, to the limited number of reagents for visualizing presynaptic calcium channels and the fact that active zones in most areas of the nervous system are small structures, at or near the diffraction limit of light. Although auxiliary calcium channel subunits such as α2δ have mutant phenotypes that affect presynaptic calcium influx and calcium channel abundance (*Dickman et al., 2008*; *Ly et al., 2008*; *Hoppa et al., 2012*), it remains unknown whether these auxiliary subunits acutely modulate calcium channel abundance in response to activity, as opposed to simply establishing and maintaining baseline expression levels during development and throughout the life of a synapse. As such, exploration of how calcium channel abundance is modulated during PHD should be of considerable interest.

It is interesting to note that while our measurements of presynaptic calcium channel abundance are restricted to the active zone, defined by the presence of Brp labeling, we did not observe an increase in non-synaptic CaV2.1-GFP signal. Furthermore, because we measured CaV2.1-GFP driven from a transgene under control of the GAL4-UAS expression system, whatever mechanisms are at play during PHD likely act downstream of calcium channel transcription. Therefore, we hypothesize that PHD is driven either by restricting CaV2.1 abundance at the level of translation, or by the regulated internalization and degradation of presynaptic CaV2.1 calcium channels.

### Calcium channel number, vesicle release and the existence of additional presynaptic mechanisms of PHD

We have documented a dramatic decrease in presynaptic calcium channel abundance during PHD, approaching a 50% decrease in total active-zone associated CaV2.1-GFP levels. We acknowledge that we are visualizing transgenically over-expressed channels, tagged with GFP, rather than the

endogenous protein. However, the documented ~50% decrease in CaV2.1-GFP abundance is mirrored, quantitatively, by a ~50% drop in the spatially averaged presynaptic calcium signal in response to single action potential stimulation. Furthermore, baseline neurotransmitter release is unaltered by CaV2.1-GFP overexpression, consistent with prior observations (*Kawasaki et al., 2004*; *Wang et al., 2014*), suggesting that the CaV2.1-GFP containing channels replace endogenous, untagged channels with little or no change in total calcium channel number at the active zone. Based upon these considerations, we argue that our measurements of decreased calcium channel number and decreased calcium influx are an accurate reflection of what is happening at the active zone during PHD.

We are also confident of our estimation of presynaptic quantal content in *vGlut*-OE animals. There is a ~50–60% increase in quantal size that is precisely offset by a ~30% decrease in presynaptic vesicle release (quantal content, *Figure 1*). These measurements are consistent with the originally described phenotype of *vGlut*-OE animals (*Daniels et al., 2004*). Furthermore, we have replicated and extended the finding that PHD is robustly expressed over a range of extracellular calcium concentrations (0.3–3 mM).

Even though our measurements of calcium channel number and calcium influx are internally consistent, and our measurements of quantal content during PHD corroborate prior publications, there remains a discrepancy. The magnitude of the decrease in calcium channel abundance and presynaptic calcium influx should cause a much larger decrease in quantal content that what we observe during PHD. More specifically, the relationship between release and external calcium is a power function with an exponent of approximately three at the *Drosophila* NMJ (*Dickman and Davis, 2009*; *Müller et al., 2011*). Therefore, a 50% drop in the spatially averaged calcium signal should decrease release by more than the observed 30%. The reason for this discrepancy remains unknown. We recently documented a linear relationship between the RRP and extracellular calcium at this synapse (*Müller et al., 2015*), consistent with similar observations at the mammalian calyx of held (*Thanawala and Regehr, 2013*). One possibility, therefore, is that the mechanisms of PHD include two inter-dependent processes. First, there is a drop in calcium channel abundance. Second, there is a mechanism that potentiates the RRP to achieve precise homeostatic compensation. This hypothetical, calcium-independent, potentiation of the RRP would offset the expected reduction in the RRP attributable to decreased calcium influx and result in an RRP measurement that does not appear to change during PHD. If this model is correct, it should be possible to isolate mutations that interfere with the latter mechanism and cause presynaptic release to drop below the levels normally seen for precise PHD. This is not observed when a *rim* mutation is placed in the background of *vGlut*-OE animals, indicating that any such mechanism must be independent of the activity of RIM.

Alternatively, recent evidence indicates that increasing vesicular size may directly increase $P_r$ at hippocampal synapses (*Herman et al., 2014*), suggesting another model. In this model, the overexpression of *vGlut2*, which causes increased vesicle diameter (*Daniels et al., 2004*), would cause a concomitant increase in $P_r$. Because this increase in $P_r$ would occur in addition to increased quantal size, one would expect a much larger compensatory response as compared to perturbing quantal size alone, therefore explaining the apparent outsized reduction in calcium influx. This model would also explain why only a mild reduction in $P_{train}$ is observed electrophysiologically at *vGlut-OE* synapses, despite the large decrease in presynaptic calcium influx. Further exploration of PHD should clarify whether *vGlut2* overexpression itself increases $P_r$ and/or whether additional mechanisms exist to oppose CaV2.1 down-regulation.

## Implications from the combined, accurate expression of PHP and PHD

Application of PhTx to *vGlut*-OE animals results in an NMJ that releases enlarged vesicles, but with resulting mEPSPs that are smaller than normal. In the presence of both perturbations, the homeostatic signaling system is able to restore EPSP amplitudes to precisely wild type levels. Because PhTx induces homeostatic potentiation in 10 min, and because there does not appear to be a reservoir of CaV2.1 channels present extra-synaptically in *vGlut*-OE animals, it is unlikely that there is sufficient time to repopulate all of the active zones at the NMJ with calcium channels, thereby erasing PHD and allowing the system to function solely according to PHP. Thus, the correct restoration of EPSP amplitude likely occurs through the summation of PHP and PHD. In this model, the overexpression of *vGlut* induces PHD and establishes a new steady state marked by increased mEPSP amplitude and decreased calcium channel abundance, with subsequent PhTx application independently inducing

PHP through trafficking of ENaC channels to the presynaptic membrane. This model predicts that the opposite order of perturbation, early induction of PHP with a subsequent acute induction of PHD, would have a similar outcome, though the reagents to test this possibility do not yet exist.

Both PHP and PHD presumably require retrograde signaling from the muscle to the motor neuron to drive the observed changes in presynaptic release. Diverse retrograde, trans-synaptic signaling systems have been observed including endocannabinoids at mammalian synapses (*Kreitzer and Regehr, 2001*) and other forms of target-derived signaling at *Aplysia* sensory-motor synapses (*Bao et al., 1998*; *Antonov et al., 2003*; *Hu et al., 2006*; *Cai et al., 2008*). A unique aspect of homeostatic plasticity is that retrograde signaling accurately and persistently adjusts presynaptic release to offset a perturbation. As such, the retrograde signaling system(s) seem to have the capacity to convey quantitative information regarding the magnitude of the postsynaptic perturbation (*Davis, 2013*). At the *Drosophila* NMJ, recent evidence implicates Endostatin as a trans-synaptic signaling molecule necessary for homeostatic plasticity (*Wang et al., 2014*), but little else is known. Ultimately, identifying the retrograde signals responsible for PHP and PHD and defining how these signals interact, will be necessary to understand how homeostatic, rheostat-like control of presynaptic release is achieved.

In conclusion, our data define a previously unknown layer of complexity driving presynaptic homeostatic plasticity, with parallel pathways acting to correct bi-directional perturbation. Moreover, we identify a novel mechanism of synaptic plasticity that is driven by tight regulation of presynaptic calcium channel abundance. Given the importance of voltage-gated calcium channels in controlling synaptic transmission, as well as the numerous human diseases linked to their dysfunction (*Ophoff et al., 1996*; *Zhuchenko et al., 1997*; *Striessnig et al., 2010*), further investigations into the mechanisms of CaV2.1 regulation during PHD should be of broad relevance.

## Materials and methods

### Fly stocks and genetics

*Drosophila* stocks were maintained at 22–25°C on normal food. The $w^{1118}$ strain was used as a wild-type control, matching the genetic background of all transgenic lines used in this study. *UAS-vGlut2* flies were obtained from Aaron DiAntonio. In all cases, *vGlut2* overexpression was driven by *OK371-Gal4*. *rim$^{103}$* mutant animals were previously generated (*Müller et al., 2012*). A plasmid containing Archaerhodopsin3 (Arch) with a C terminal GFP tag was obtained (Addgene plasmid 22217). The coding sequence for Arch was cloned into the *pUAST* destination vector to generate *UAS-Arch*. This construct was confirmed by sequencing. Transgenic flies were generated using standard injection methods by BestGene Inc. Stocks containing *UAS-Arch* insertions on chromosome two and three were used for experiments. All imaging and electrophysiology experiments on Arch expressing flies were performed using two copies of *UAS-Arch* (2 on chromosome II, or 1 on II and 1 on III) and one copy of the motor neuron driver *OK371-GAL4*. For Arch imaging experiments, crosses were set up and allowed to lay for 2–3 days on food containing 1 mM *all-trans* retinal (ATR). Crosses containing ATR food were wrapped in foil and kept at 25°C. All other flies were obtained from the Bloomington *Drosophila* Stock Center.

### Electrophysiology

Sharp-electrode recordings were made from muscle six in abdominal segments two and three of third instar larvae using an Axopatch 200B, or a Multiclamp 700B amplifier (Axon Instruments), as previously described (*Davis and Goodman, 1998*). Two-electrode voltage clamp recordings were performed with an Axoclamp 2B amplifier. The extracellular HL3 saline contained (in mM): 70 NaCl, 5 KCl, 10 MgCl$_2$, 10 NaHCO$_3$, 115 sucrose, 4.2 trehalose, 5 HEPES, and various concentrations of CaCl$_2$ (see 'Results' and Figures). To assess PHP, larvae were incubated in Philanthotoxin-433 (PhTX; 20 μM; Sigma–Aldrich, St. Louis, MO) for 10 min (*Frank et al., 2006*). The average single AP-evoked EPSP amplitude (stimulus duration, 3 ms), or EPSC amplitude (stimulus duration, 1 ms) of each recording is based on 30 presynaptic stimuli. Quantal content is given by the ratio of the average EPSP amplitude over average mEPSP amplitude of a recording, and then averaging recordings across all NMJ for a given genotype. The apparent size of the readily-releasable vesicle pool (RRP) was probed by the method of cumulative EPSC amplitudes (*Schneggenburger et al., 1999*), which was recently applied to the *Drosophila* NMJ (*Hallermann et al., 2010*; *Miśkiewicz et al., 2011*; *Weyhersmüller et al., 2011*; *Müller et al., 2012*). Muscles were clamped to their resting membrane potential (Vm), or

clamped to −65 mV if the Vm was more positive than −60 mV. Muscles with a Vm more depolarized than −55 mV were discarded. Synapses were stimulated with 60-Hz trains (30 stimuli, at least five trains per synapse). EPSC amplitudes during a stimulus train were calculated as the difference between peak and baseline before stimulus onset of a given EPSC. The cumulative EPSC amplitude was obtained by back-extrapolating a line fit to the last 10 cumulative EPSC amplitude values of the 60 Hz train to time 0. Electrophysiology data were acquired with Clampex (Axon Instruments, Foster City, CA), and imaging data were recorded with Prairie View (Prairie Technologies, Middleton, WI). Data were analyzed with custom-written routines using Igor Pro 6.2.2. (Wavemetrics; custom script submitted with this manuscript), and mEPSPs were analyzed with Mini Analysis 6.0.3. (Synaptosoft, Decatur, GA). Statistical significance was assessed with a Student's t-test, and all error bars are SEM.

## Ca²⁺ imaging

Ca$^{2+}$ imaging experiments were done as described in *Müller and Davis (2012)*. Third instar larvae were dissected and incubated in ice cold, Ca$^{2+}$-free HL3 containing 5 mM Oregon-Green 488 BAPTA-1 (OGB-1; hexapotassium salt, Invitrogen) and 1 mM Alexa 568 (Invitrogen). After incubation for 10 min, the preparation was washed with ice cold HL3 for 10–15 min. This leads to an intraterminal OGB-1 concentration of approximately 50 μM (*Müller and Davis, 2012*). Single action-potential evoked spatially-averaged Ca$^{2+}$ transients were measured in type-1b boutons synapsing onto muscle 6/7 of abdominal segments A2/A3 at an extracellular [Ca$^{2+}$] of 1 mM using a confocal laser-scanning system (Ultima, Prairie Technologies) at room temperature. Excitation light (488 nm) from an air-cooled krypton-argon laser was focused onto the specimen using a 60× objective (1.0 NA, Olympus), and emitted light was detected with a gallium arsenide phosphide-based photocathode photomultiplier tube (Hamamatsu). Line scans across single boutons were made at a frequency of 313 Hz. Fluorescence changes were quantified as $\Delta F/F = (F(t) - F_{baseline})/(F_{baseline}-F_{background})$, where F(t) is the fluorescence in a region of interest (ROI) containing a bouton at any given time, Fbaseline is the mean fluorescence from a 300-ms period preceding the stimulus, and Fbackground is the background fluorescence from an adjacent ROI without any indicator-containing cellular structures. One synapse (4–12 boutons) was imaged per preparation. The average Ca$^{2+}$ transient of a single bouton is based on 8–12 line scans. Experiments in which the resting fluorescence decreased by > 15%, and/or which had a Fbaseline > 650 a.u. were excluded from analysis. Data of experimental and control groups were collected side by side. The Ca$^{2+}$ indicator was not saturated by single AP stimulation because repetitive stimulation induced a further increase in the peak ΔF/F amplitude.

## Arch imaging

Confocal spot imaging was made from type 1b boutons on muscle 6/7 of abdominal segments 2–4 of third instar larvae using a confocal laser-scanning microscope (Ultima, Prairie Technologies). Excitation light (643 nm, 4 mW at back of objective) from an air-cooled solid-state laser was focused onto the specimen using a 60× objective (1.0 NA, Olympus). The emission path consisted of a quad band 405, 488, 515, 643 nm dichroic, 500 nm long pass filter, 600 nm dichroic, and 700/75 nm and 525/40 nm band pass emission filters for Arch and GFP emissions, respectively. Arch emission was detected with a gallium arsenide phosphide-based photocathode photomultiplier tube (Hamamatsu) and GFP emission was detected by a second PMT. Spot imaging from edges of single boutons was made at a sampling frequency of 4 kHz. A 1 ms stimulation of the nerve fiber was used to evoke action potentials after 60 ms of baseline image acquisition. 30–50 events were collected for each bouton at an inter-stimulus rate of 0.5 Hz for single action potentials and an inter-train interval of 0.1 Hz for trains of five stimuli. 1–5 boutons per synapse were imaged. For experiments using 1.5 mM calcium, 10 μM Philanthotoxin-433 (Sigma) or 100 μM 1-Naphthylacetyl spermine (Sigma) was included in the saline to prevent contraction during stimulation. Imaging data were analyzed using custom-written routines in Matlab (Mathworks; custom script submitted with this manuscript) and digitally filtered at 2 kHz. Fluorescence signals consisted of an exponentially decaying signal that derives from tissue fluorescence and is independent of Arch fluorescence, a rapid increase in Arch fluorescence that occurs during the first 10 ms of the photocycle (*Maclaurin et al., 2013*), and the voltage dependent change in Arch fluorescence. To isolate voltage dependent changes in fluorescence, we fit a single exponential from 10 ms after the start of imaging to 5 ms before stimulus onset. This fit, which approximates the tissue fluorescence as well as the baseline Arch fluorescence, was extrapolated and subtracted from the fluorescence. Traces containing extra action potentials were removed.

The resulting fluorescence measurements were averaged for each bouton and the average waveform was used to determine peak amplitude, full width, and width at half maximum (WHM). To obtain precise width and WHM measurements, we used linear interpolation between data sampling points. Due to the inability to accurately measure baseline Arch fluorescence in the presence of tissue fluorescence and Arch localized to internal membranes, amplitude measurements are not reported. Average AP waveforms for each experimental condition were peak-aligned and normalized before averaging.

### Immunocytochemistry

Standard immunocytochemistry was performed as previously described. Dissected third instar larvae were fixed with ice-cold ethanol for 5′. The following primary antibodies were used: mouse anti-Brp (1:100) (*Kittel et al., 2006*), and rabbit anti-GFP (1:1,000, Invitrogen clone 3E6). Alexa-conjugated secondary antibodies were used at 1:300 (Jackson Immuno-research Laboratories).

### Acknowledgements

The authors declare no competing financial interests. We thank Özgür Genç and TingTing Wang for comments on the manuscript. This work was supported by NIH Grant number SN079307 to GWD and F32 NS086164-02 to MAG.

## Additional information

### Competing interests

GWD: Reviewing editor, *eLife*. The other authors declare that no competing interests exist.

### Funding

| Funder | Grant reference | Author |
| --- | --- | --- |
| National Institutes of Health | SN079307 | Graeme W Davis |
| National Institutes of Health | F32 NS086164-02 | Michael A Gaviño |

The funders had no role in study design, data collection and interpretation, or the decision to submit the work for publication.

### Author contributions

MAG, Conception and design, Acquisition of data, Analysis and interpretation of data, Drafting or revising the article, Contributed unpublished essential data or reagents; KJF, SA, Acquisition of data, Analysis and interpretation of data; GWD, Conception and design, Analysis and interpretation of data, Drafting or revising the article

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
