## [Decision Letter]

Thank you for sending your work entitled “Homeostatic Synaptic Depression is Achieved Through a Regulated Decrease in Presynaptic Calcium Channel Abundance” for consideration at *eLife*. Your article has been favorably evaluated by Eve Marder (Senior editor) and three reviewers, one of whom is a member of our Board of Reviewing Editors.

The following individuals responsible for the peer review of your submission have agreed to reveal their identity: Ronald L Calabrese (Reviewing editor and reviewer), David Glanzman (reviewer), and Troy Littleton (reviewer).

The Reviewing editor and the other reviewers discussed their comments before we reached this decision, and the Reviewing editor has assembled the following comments to help you prepare a revised submission.

The authors present very thorough analyses of PHD and its interaction with PHP (bidirectional homeostatic plasticity mechanisms) at the *Drosophila* NMJ. Here the authors' provide evidence that 1) homeostatic processes that depress or enhance (PHD and PHP, respectively) synaptic function at *Drosophila* NMJs via genetically separable mechanisms, and 2) that PHD occurs by decreased calcium influx via reduction in calcium channels (CaV2.1) at individual active zones. The experiments employ state of the art electrophysiological, imaging, genetic, and morphological labeling techniques. The observations that the two forms of homeostasis occur via independent mechanisms and can be co-expressed (at least with genetic and pharmacological manipulations) are quite intriguing. The figures are very clear and show necessary data. The writing is crisp and clear. This paper should provoke wide interest among the readers of *eLife*.

The reviewers have discussed the manuscript and are all highly enthusiastic about the work. They have identified some small issues that the authors should address prior to publication and which can be re-reviewed by the Reviewing editor.

1) The authors report in Figure 7 that overexpressed Cac-GFP levels at individual active zones are reduced in *vGlut-*OE lines compared to WT. From this experiment, the authors propose that PHD expression occurs via reduction in the number of calcium channels at individual active zones. Can the authors also report the number of active zones (immunocytochemical counts of active zones numbers with an anti-BRP antibody) and Cac/GFP puncta)/bouton as well? From the representative images, the *vGlut*-OE lines appear to shown a reduced number of active zones per bouton, something that could affect the expression mechanisms of PHD. Excluding a possible reduction in the number of active zones per bouton could more strongly support the author's conclusion that expression of PHD is mediated by reduction in calcium channel number per active zone. Including quantification for synapse number (boutons and active zones) in *vGlut*-OE lines compared to WT would provide some insights into potential structural alterations in synapses that might accompany (and potentially modify) the types of plasticity observed.

2) One important question that these results do not address is the possible postsynaptic mechanisms that contribute to PHP and PHD. In the case of PHP, in particular, there must be some postsynaptic signal triggered by the blockade of glutamate receptors in the motor neuron. How does this postsynaptic signal cause the presynaptic changes that lead to increased presynaptic release in the case of PHP? Moreover, it is possible, indeed likely, that there is a postsynaptic mechanism that senses the increase in mini amplitude produced by overexpression of presynaptic v*Glut2*; this postsynaptic sensor might produce a retrograde signal that causes the decrease in the number of CaV2.1 channels in the presynaptic terminals observed in the present study. There is ample evidence for regulation of presynaptic mechanisms of plasticity by retrograde signaling at other synapses, e.g., the *Aplysia* sensory-motor synapse. It is disappointing that the authors do not discuss potential postsynaptic contributions to PHP and PHD in their paper. To my mind, this leaves a large conceptual gap in their scheme of how these two forms of homeostatic plasticity are independently regulated at NMJs.

3) The authors provide experiments in Figure 3 suggesting that distinct mechanisms mediate PHD and PHP. The experiments indicate PHD is maintained in 3 genetic or pharmacological manipulations that have been previously shown to block PHP. The conclusions could be strengthened if the authors performed experiments demonstrating that PHP was indeed blocked in at least one of these preparations. For example, is PTHX induced PHP blocked in *rim*^*103*^ + *vGlut*-OE animals that express PHD as predicted? Inclusion of this control, while not absolutely necessary would more strongly support the authors' conclusion that PHD and PHP occur via experimentally separable mechanisms.

---

## [Author Response]

*1) The authors report in*
Figure 7
*that overexpressed Cac-GFP levels at individual active zones are reduced in vGlut-OE lines compared to WT. From this experiment, the authors propose that PHD expression occurs via reduction in the number of calcium channels at individual active zones. Can the authors also report the number of active zones (immunocytochemical counts of active zones numbers with an anti-BRP antibody) and Cac/GFP puncta)/bouton as well? From the representative images, the vGlut-OE lines appear to shown a reduced number of active zones per bouton, something that could affect the expression mechanisms of PHD. Excluding a possible reduction in the number of active zones per bouton could more strongly support the author's conclusion that expression of PHD is mediated by reduction in calcium channel number per active zone. Including quantification for synapse number (boutons and active zones) in vGlut-OE lines compared to WT would provide some insights into potential structural alterations in synapses that might accompany (and potentially modify) the types of plasticity observed*.

This is an excellent point. We have now quantified the number of active zones per bouton. There are no statistically significant changes in vGlut-OE animals compared to wild type controls. These values are provided in the text including samples sizes and the results of our statistical tests.

*2) One important question that these results do not address is the possible postsynaptic mechanisms that contribute to PHP and PHD. In the case of PHP, in particular, there must be some postsynaptic signal triggered by the blockade of glutamate receptors in the motor neuron. How does this postsynaptic signal cause the presynaptic changes that lead to increased presynaptic release in the case of PHP? Moreover, it is possible, indeed likely, that there is a postsynaptic mechanism that senses the increase in mini amplitude produced by overexpression of presynaptic vGlut2; this postsynaptic sensor might produce a retrograde signal that causes the decrease in the number of CaV2.1 channels in the presynaptic terminals observed in the present study. There is ample evidence for regulation of presynaptic mechanisms of plasticity by retrograde signaling at other synapses, e.g., the Aplysia sensory-motor synapse. It is disappointing that the authors do not discuss potential postsynaptic contributions to PHP and PHD in their paper. To my mind, this leaves a large conceptual gap in their scheme of how these two forms of homeostatic plasticity are independently regulated at NMJs*.

This is an interesting point and one that we are more than happy to discuss, as we have been trying to identify the molecular nature of retrograde signaling systems at our system for many years. We have added a new paragraph to the Discussion section of our text. In this paragraph we cite the potential requirement for retrograde signaling systems and we cite the outstanding work that has been performed previously at mammalian synapses with the discovery of retrograde endocanabinoid signaling as well as work performed at the Aplysia sensory-motor synapse. Truly, this is the subject for a more extensive review. But, we hope that by raising this issue we have adequately addressed this reviewer’s concern.

*3) The authors provide experiments in*
Figure 3
*suggesting that distinct mechanisms mediate PHD and PHP. The experiments indicate PHD is maintained in 3 genetic or pharmacological manipulations that have been previously shown to block PHP. The conclusions could be strengthened if the authors performed experiments demonstrating that PHP was indeed blocked in at least one of these preparations. For example, is PTHX induced PHP blocked in rim*^*103*^
*+ vGlut-OE animals that express PHD as predicted? Inclusion of this control, while not absolutely necessary would more strongly support the authors' conclusion that PHD and PHP occur via experimentally separable mechanisms*.

We now provide a control in which we demonstrate that application of the ENaC inhibitor Benzamil blocks PhTx-driven homeostatic potentiation in the vGlut-OE background. We chose this experiment as it also supports our experimental model for the integration of PHP and PHD, which proposes that new ENaC channels will still be sufficient to potentiate calcium influx even though calcium channel number is diminished in the vGlut-OE background.